# A Review on Membrane Biofouling: Prediction, Characterization, and Mitigation

**DOI:** 10.3390/membranes12121271

**Published:** 2022-12-15

**Authors:** Nour AlSawaftah, Waad Abuwatfa, Naif Darwish, Ghaleb A. Husseini

**Affiliations:** 1Department of Chemical and Biological Engineering, College of Engineering, American University of Sharjah, Sharjah P.O. Box 26666, United Arab Emirates; 2Materials Science and Engineering Program, College of Arts and Sciences, American University of Sharjah, Sharjah P.O. Box 26666, United Arab Emirates

**Keywords:** biofouling, biofilm, biocides, artificial intelligence, biofouling mitigation

## Abstract

Water scarcity is an increasing problem on every continent, which instigated the search for novel ways to provide clean water suitable for human use; one such way is desalination. Desalination refers to the process of purifying salts and contaminants to produce water suitable for domestic and industrial applications. Due to the high costs and energy consumption associated with some desalination techniques, membrane-based technologies have emerged as a promising alternative water treatment, due to their high energy efficiency, operational simplicity, and lower cost. However, membrane fouling is a major challenge to membrane-based separation as it has detrimental effects on the membrane’s performance and integrity. Based on the type of accumulated foulants, fouling can be classified into particulate, organic, inorganic, and biofouling. Biofouling is considered the most problematic among the four fouling categories. Therefore, proper characterization and prediction of biofouling are essential for creating efficient control and mitigation strategies to minimize the damage associated with biofouling. Moreover, the use of artificial intelligence (AI) in predicting membrane fouling has garnered a great deal of attention due to its adaptive capability and prediction accuracy. This paper presents an overview of the membrane biofouling mechanisms, characterization techniques, and predictive methods with a focus on AI-based techniques, and mitigation strategies.

## 1. Introduction

Clean water is a finite resource with continuously growing demand; according to UN-Water, around 2.3 billion people live in water-stressed countries, of which 733 million live in high and critically water-stressed countries [1]. Desalination provides a way to produce fresh water more suitable for human use and agriculture from saline or brackish water [2]. Due to reductions in production costs and time, membrane-based separation in water treatment has gained increased popularity over the past few decades. Commonly used membrane separation technologies include reverse osmosis (RO), micro- and ultrafiltration, membrane distillation (MD), and electrodialysis (ED), while membrane crystallization (MCr) and pervaporation (PV) are still under laboratory evaluation [2,3,4].

Membrane technology has been extensively evaluated for water desalination. With respect to economic analyses of the costs and energy consumption associated with these methods, Nthunya et al. [3] provided a comprehensive review in which they compared the capital and operating expenditures (CAPEX and OPEX) of RO, ED, MD, and MCr for different-sized desalination processes of brackish water and seawater. In their analysis, capital costs encompassed design costs, transport, equipment, project management, instrumentation, infrastructure, and buildings. In comparison, operation, and maintenance (O&M) costs included labor, insurance, energy, maintenance, and consumables. Their findings showed that O&M costs were lower for ED compared to the other membrane techniques investigated, and the small-scale MD OPEX was lower than that for RO. However, there is not much reported on the large-scale O&M of MD due to its slow industrial growth. Accordingly, they concluded that RO is the most preferrable membrane separation process in the current water desalination markets. Kesieme et al. [2] conducted an economic analysis comparing MD and RO. Considering a reference 30,000 m^3^/day plant, RO was economically favorable even with the inclusion of the carbon tax ($23 per ton carbon) in Australia. However, if heat is available at low costs, the cost of MD would have been reduced to $0.66/m^3^, which is cheaper than RO. The authors raised an important point regarding carbon emissions taxes, stating that with policies coming into practice to tax carbon emissions, the economics of these membrane processes will undergo changes that introduce uncertainties in costing reports as to what desalinated water will cost in a carbon-constrained society.

In membrane processes, the membrane is always in contact with the solutions being treated. Consequently, it is prone to chemical or biological deposition of matter. Membrane fouling refers to the deposition and accumulation of materials on or in the membrane. It results from complex interactions between the various foulants in the feed and the membrane surface [5]. There are four types of fouling, namely, complete pore blocking, partial pore blocking, internal pore blocking, and cake formation (Figure 1). In order to control these types, the nature of the foulants present must be known; accordingly, membrane foulants can be categorized into particulate, organic, inorganic, and biofoulants [6,7,8,9,10,11].

Biofouling, which involves the accumulation of biological microorganisms followed by the formation of a biofilm on the membrane [6,12], can be divided into microfouling and macrofouling. Microfouling involves the accumulation of unicellular or multicellular microorganisms (e.g., bacteria, yeast, or fungi) that may form a biofilm by mono-species or multi-species, whereas macrofouling is associated with bigger organisms, such as algae [10]. Biofouling accounts for approximately 45% of membrane fouling and is generally regarded as the most problematic among the four fouling categories [5]. 

The topic of membrane fouling has been investigated intensely in literature, with numerous reviews having been written on the topic. Some of these reviews are focused on the general aspects of membrane fouling. For example, Guo and Ngo [9] identified the major foulants, the principal membrane fouling mechanisms, as well as possible mitigation processes. Charcosset [11] reviewed the main characteristics of membrane processes, membrane fouling, energy consumption, and associated environmental issues. In comparison, Rudolph et al. [13] presented a review of the state-of-the-art techniques used for in situ membrane monitoring. In other reviews, membrane fouling was examined in relation to a particular membrane separation method; for instance, Shi et al. [14] reviewed the different techniques available for predicting fouling in membrane bioreactors. Qasim et al. [4] provided an extensive review of the theories and models underlying membrane transport, and membrane fouling. They also provided a thorough discussion of the different membrane cleaning and pretreatment technologies, in addition to current challenges faced by RO membrane processes. Hubadillah et al. [15] reported on alternative techniques to RO, such as forward osmosis (FO) and MD. With respect to artificial intelligence (AI), Bagheri et al. [16], Lim et al. [17], in addition to Viet and Jang [18], presented a general overview of the application of AI to membrane fouling prediction. 

Although there are a few reviews that have delved into the topic of membrane biofouling [19,20,21], there is still a gap in the literature when it comes to reviews focused solely on the anti-biofouling mechanism and mitigation strategies. Therefore, the objective of this article is to introduce the mechanism of biofouling and existing characterization techniques, while building on the information provided by these previous reviews. Moreover, the discussion of the prediction of membrane biofouling in this review focuses more on AI-based prediction models. Finally, membrane biofouling control strategies based on recent research progress are detailed, as well as a description of some new studies in this area.

## 2. Membrane Biofouling Mechanism

Membrane biofouling occurs in three main steps: attachment, propagation, and biofilm formation. The first step involves the deposition and physical adsorption of microorganisms (algae, protozoa, bacteria, and fungi) to the membrane surface. This process is controlled by three main factors [22,23,24,25]: Microorganism: this includes species, population density, growth profile, nutrient status, the hydrophobicity/hydrophilicity of the microorganism, and physiological responses.Surface morphology: membrane material, surface charge, hydrophobicity, roughness, and porosity.Feed: temperature, pH, dissolved organic/inorganic matter, shear forces, and flux.

Hydrodynamic forces or physicochemical interactions can drive the adhesion of microorganisms to the surface. The convective nature of hydrodynamic forces brings suspended microorganisms and other foulants close to the membrane surface, facilitating their adhesion to the surface. Physicochemical interactions can be divided into long-range Lifshitz–van der Waals interactions, and short-range Lewis acid–base and electrostatic double-layer interactions [6,21]. Adsorption is followed by cell growth and multiplication [19,22]. During the colonization and proliferation stage, the attached microorganisms secrete a polymeric material known as the extracellular polymeric substance (EPS) matrix, which further anchors them to the membrane [23] (refer to Figure 2). The EPS matrix composition is highly dependent upon the environment in which the biofilm develops; however, EPS matrices usually contain organic molecules (e.g., proteins, nucleic acids, lipids, and polysaccharides), and inorganic matter (e.g., minerals, clay, and corrosion products). Among the EPS matrix components, proteins play a significant role in biofouling because they provide an optimum environment for microbial colonization, i.e., the building blocks of proteins (amino acids) possess several functional groups, such as carboxyl, amino, and methyl groups. The presence of these functional groups affects the hydrophilicity of proteins, which in turn influences the adhesion of the EPS-enclosed microorganisms to the membrane through diverse intramolecular forces, such as van der Waals forces, hydrogen bonding, and hydrophobic interactions [6,8,10,26,27]. 

The impact of biofouling on the membrane processes includes flux decline, damage to the membrane structure, inhibition of conventional transport mechanisms, increased feed and differential pressure, increased energy consumption, and the need for frequent cleaning, which adversely affects membrane plant operation and shortens membrane life [22,28]. 

## 3. Membrane Biofouling Characterization

The characterization of membrane biofouling involves evaluating the microbial community, the fouling layer, and the quantification of the EPS matrix. Various assays, microscopic and spectroscopic techniques have been employed to characterize membrane biofouling [29]. Kerdi et al. [30] used 3D-optical coherence tomography (OCT) to characterize the intrinsic structure and the mechanical properties of the biofilm developing on ultrafiltration (UF) polyethersulfone membranes without altering their chemical and/or physical properties. Three-dimensional images of the biofilm were obtained with high resolution, enabling the biofilm microstructural morphology analysis. The structural properties were found to be dependent of time as the biofilm continuously evolved, i.e., the biofilm was more elastic in nature at the initial stages of its growth. Still, it then transitioned into a more viscoelastic type as it matured. Benladghem et al. [31] used surface enhanced Raman spectroscopy (SERS) to identify biofoulants accumulated on spiral-wound reverse osmosis (SWRO) membranes. They also imaged the fouled membranes’ topography and the biofilm structures using fluorescence microscopy (FM) and scanning electron microscopy (SEM). The microscopy images showed that both biotic and abiotic deposits were present on the membrane; in addition, SERS showed that the thickness of the fouling layer reached up to 5µm. In another study, Zahid et al. [32] used thermal analysis (differential scanning calorimetry, DSC), contact angle, SEM, Fourier transform infrared (FTIR) spectroscopy, and the disc diffusion method to characterize cellulose acetate (CA)-sulfonated graphene oxide (SGO)-doped membranes and assess their anti-biofouling properties. The functionalization of CA membranes with SGO was confirmed using FTIR, while the morphology of the doped membranes was studied using SEM. The thermal stability of the doped membranes was investigated using DSC and revealed an increase in thermal stability of CA-SGO membranes upon the addition of SGO. *Escherichia coli* pathogenic bacteria were used in the disc/agar well diffusion test through measurement of inhibition zone. The test results showed that that antibacterial activity of the CA membrane is increased with increased SGO content of SGO. This behavior was attributed to the fact that the surface of CA-SGO-doped membranes had a negative charge from the sulfonic acid and hydroxyl groups of SGO, which led to electrostatic repulsion between the microorganisms and the membrane. Masigol et al. [33] developed an interesting technique to characterize and identify multi-species biofilms termed polymer surface dissection (PSD). PSD uses targeted and ultraviolet (UV)-responsive polyethylene glycol hydrogels to bind to and detach microorganism aggregates from membranes. The detached aggregates are then exposed to UV light to release aggregates of the desired size for DNA extraction. The efficacy of PSD was demonstrated by identifying the bacterial community structure (aggregate area of 5000−60,000 μm^2^) developed during early-stage biofouling of aerobic wastewater communities over polyvinylidene difluoride (PVDF) membranes. The findings showed that larger aggregates had less bacterial diversity. Moreover, the bacterial community structure shifted from one rich in Bacteroides to one with more proteobacteria when the aggregates areas reached the 25,000–45,000 μm^2^ size range. The efficiency of bacterial transfer between the membrane surface and the hydrogel still poses an issue for the developed method. The authors proposed using poly-Llysine (PLL) as a targeting ligand to improve transfer efficiency, as well as the application of external electric fields to enhance transfer efficiency. It is important to note that for EPS matrix characterization, the EPS matrix needs to be extracted first, which can be done using physical methods (e.g., centrifugation, dialysis, filtration, ion exchange, heating), chemical methods (commonly used chemicals include, sodium hydroxide, formaldehyde, ethanol) or a combination of both [5,23,34]. Table 1 provides a summary of these techniques and the information that can be obtained from them. 

## 4. Membrane Biofouling Prediction

Membrane fouling is an inevitable aspect of membrane operations; however, real-time, fast, and accurate predictions of membrane fouling can enhance its control, improve the efficiency of membrane operations as well as drastically reduce the involved operating costs [14,35,36]. The tactics used to predict membrane fouling have been extensively reviewed and numerous articles have been published on the topic [4,13,14,26,36,37,38,39]. Briefly, fouling prediction techniques include pilot plant evaluations, the use of fouling indices, and the use of predictive mathematical models. Some of the mathematical models developed to describe membrane fouling are listed in Table 2. Recently, the use of AI in predicting membrane fouling has garnered a great deal of attention due to its adaptive capability and prediction accuracy (refer to Figure 3) [16,40,41].

AI is an interdisciplinary field in which machines mimic human cognitive functions such as learning, problem-solving, reasoning, and perception. In simpler terms, AI can be defined as intelligence exhibited by machines whose techniques use historical data to learn about the system and adapt its decision-making processes [45,46]. They can be further divided into machine learning (ML), deep learning, and data analytics. AI algorithms such as artificial neural networks (ANN), particle swarm optimization (PSO), simulated annealing (SA), fuzzy logic (FL), adaptive neuro-fuzzy inference system (ANFIS), and support vector machine (SVM) have been applied to predict membrane fouling (refer to Table 3) [16,40,41,46,47].

Various studies demonstrated the successful application of different AI techniques for the prediction of membrane biofouling; for instance, Yokoyama et al. [48] combined NMR spectroscopy and several ML models to predict the maximum transmembrane pressure (TMP), analyze the chemical compounds causing fouling based on a chemometric analysis of NMR spectra, as well as determining their effects on fouling progress. Out of the tested models, random forest (RF) exhibited the highest accuracy in the analysis of the NMR spectra; in addition, the analysis revealed that among the bacterial-EPS components, polysaccharides contributed the most to membrane biofouling. The authors attributed this to the fact that the high molecular weight and viscosity of polysaccharides make it easier to attach them to membranes and clog pores. Qamar et al. [49] used deep neural networks (DNN) to monitor biofilm growth and connect its thickness with hydrodynamic parameters. OCT scans were used to generate a biofilm thickness database, which was then used to train the convolution neural network (CNN). The trained CNN network was able to predict biofilm thickness for different filtration technologies (mainly UF and membrane distillation) with a mean squared error (MSE) of less than 0.008 µm^2^ for a set of 300 testing images. Moreover, the ability of the non-linear-DNN to predict and relate pressure drop with biofilm growth was validated against the analytical solution with an absolute error <2%. Additional studies are presented in Table 4.

Despite the reported successes of AI in predicting membrane biofouling, some concerns have been raised regarding the use of these non-mechanistic modeling tools to correlate operating variables with performance parameters. The main source of concern is that these techniques are not based on physical or chemical phenomena and rely on calibrations ‘learning’ using experimental data. In addition, if there are any sudden changes in operating parameters, the models may be susceptible to overfitting, and misleading correlations. The selection of the proper modeling tool is also very important for the predictive accuracy of the AI tools. These issues can be overcome by simplifying model structure, optimizing input parameters, ensuring that the data sets used in the learning stage are large enough, and performing cross-calibrations across the entire calibration data set to optimize the internal structure of the algorithm and minimize data overfitting. Another criticism of AI tools is that their “black box” nature does not provide information about the physical phenomena involved in the biofouling process. On the other hand, it can be argued that the knowledge about the contribution and impact of each input in predicting outputs from the model provides mechanistic insight into the modeled processes. The use of hybrid (mechanistic and non-mechanistic) models and combining several AI models has been proposed to improve prediction accuracy and overcome the failings of a single AI model [40,46,50,51].

**Table 3 membranes-12-01271-t003:** Summary of artificial intelligence (AI) techniques commonly used in membrane fouling prediction (adapted from ([40,46,52]).

AI Technique	Mode of Operation	Applications	Advantages	Disadvantages
**k-NN**	-Saves all existing data -Classification of new data points based on similarity	-Regression -Classification	-Easy implementation	-Computationally expensive -Memory intensive -Overfitting
**DT**	-Generates a training model to teach simple decision rules	-Regression -Classification	-High accuracy -Easy implementation -Applies to continuous and discrete data	-Instability -Overfitting
**RF**	-Creates DTs on data samples -Makes predictions based on each DT -Uses a voting mechanism to select an optimal solution	-Regression -Classification	-Decreased overfitting -Suitable for large datasets	-Not suitable for imbalanced datasets -Low training speed
**ANN**	-Statistical models built based on human brain neurons	-Pattern recognition -Performs nonlinear computations	-Fast prediction -Good for arbitrary function approximation -Suitable for high-dimensional datasets	-Computationally expensive -Difficulty in interpreting trained models
**FNN**	-Combines fuzzy logic and NNs	-Pattern recognition -Density estimation -Regression -Classification	-Can be used when a mathematical model does not exist for a problem -Easy implementation and interpretation	-Theoretical knowledge necessary -Computationally expensive
**CNN, FFNN**	-Uses convolution instead of matrix multiplication	-Image/video recognition -Classification -Regression -Segmentation	-Accurate results -Good speed	-Computationally expensive -Complex architecture
**DNN**	-Input, output layers -Includes hidden layers	-Learning complex models -High-dimensional data processes	-Best performance if enough data are available -Suitable for nonlinear data -Fast prediction following training	-Computationally expensive -Requires more training data
**SVM**	-Requires labeled training data for each category to identify the next step -Mapping input vector into a high-dimensional feature space	-Classification -Regression -Pattern recognition	-Suitable for high-dimensional datasets -Suitable for linear and nonlinear datasets	-Computationally expensive -Difficult to train -Overfitting -Not suitable for noisy data
**GA**	-Produces the optimal strategy to solve complicated problems under a particular theory	-Regression -Clustering -Classification	-Provides multiple solutions -Supports multi-objective optimization -Suitable for discrete and continuous data	-Difficult to implement -Computationally expensive -Time-consuming
**PSO**	Optimizes a problem by iteratively improving a candidate solution with regard to a given measure of quality	-Clustering -Regression -Classification	-Easy implementation -Parallel computation	-Mathematical background needed for evaluations -Difficult to define initial design parameters

Abbreviations: k-NN, k-nearest neighbor; DT, Decision tree; RF, Random Forest; ANN, Artificial neural networks; FNN, Fuzzy neural networks; CNN/FFNN, Convoluted/feed-forward neural networks; DNN, Deep neural networks; SVM, Support vector machine; GA, Genetic algorithm; PSO, Particle swarm optimization.

**Table 4 membranes-12-01271-t004:** Summary of studies that used AI to predict biofouling.

Membrane Separation Process	AI Tool	Main Findings	Ref.
Extractive membrane bioreactor	ANN	-ANN was able to interpret complex 2D fluorescence maps. -Properly trained ANN was able to predict process behavior and identify key fluorophores for the prediction of process parameters.	[53]
Nanofiltration (NF)	Multivariate projection to latent structures (MPLS)	-Alkalinity, molecular size descriptors, molecular weight, and molar volume were the most relevant contributors to determining foulant rejection. -Adsorption occurred through polar and electrostatic interactions.	[54]
Ion-exchange membrane bioreactor	MPLS	-The proposed PLS model accounted for biological contribution to mass transfer. -PLS model predicted anionic fluxes across membranes with ~50% prediction improvement when compared with the simplified mechanistic Donnan dialysis-based transport model. -Transport driving force-related variables were the most important for the anionic transport model.	[55]
Osmotic membrane bioreactor	ANN	-The optimal number of hidden layers was 2–6, and the appropriate number of neurons in each layer was 5–30. -pH and conductivity were the most critical parameters for the models. -The ANN models demonstrated good performance, with R^2^ values of 0.92 and 0.93 reported for the prediction of water flux and membrane fouling simulations, respectively.	[18]
Membrane bioreactor	RF, ANN, and long-short-term memory network (LSTM)	-All models provided reliable predictions, while the RF models had the best accuracy.	[56]

## 5. Membrane Biofouling Mitigation

Despite the extensive research, no technique has been developed to completely eliminate biofouling. The main techniques used to manage biofouling are discussed in this section.

### 5.1. Feedwater Pretreatment

As mentioned earlier, feedwater composition is an essential component of membrane fouling. Feedwater pretreatment is a conventional strategy against fouling in general, where the feed is dosed with biocides, antimicrobial substances, or strong oxidants such as chlorine, hypochlorite, chlorine dioxide, ozone, or UV radiation. The efficacy of biocides and antimicrobials depends on the type of microorganisms in the system [10,23,57,58], the strength and concentration of the biocide/antimicrobial agent used [16,59], frequency of dosing, contact time, temperature, and pH in the feed water [10,16,23]. Temperature and pH are significant because they affect the growth of microorganisms; for instance, bacteria are adaptable and can colonize surfaces even at extreme conditions, such as temperatures from −12 to 110 °C and pH values between 0.5 and 13 [10,23]. Biocide treatment must be followed by high-velocity detergent cleaning and flushing to remove the organic debris, while chlorination must be followed by the addition of sodium bisulfite or by activated carbon filtration to remove any residual chlorine, which could deteriorate polymeric membranes. Chlorine dioxide has been suggested as an alternative to chlorine because it is an effective biocidal agent that is less prone to the formation of harmful by-products and has milder effects on polymeric membrane structures. However, chlorine oxide presents a handling problem since it is a gas that cannot be generated on-site [23]. Furthermore, 2,2-dibromo-3-nitrilopropionamide (DBNPA) has been proposed as a non-oxidizing biocide. The attractiveness of DBNPA lies in its fast reaction with sulfur-containing molecules in microorganisms, as well as its compatibility with polyamide membranes [60]. The main drawback of using non-oxidizing biocides (e.g., formaldehyde, glutaraldehyde) is that microbes can develop resistance to them [61].

An alternative to chemical disinfection involves the use of UV irradiation due to its ability to generate hydroxyl radicals, inhibit microbial growth, break macromolecules down into smaller fragments, as well as inactivate and destroy both bacteria and viruses (UV radiation at 254 nm can damage bacterial DNA) [62,63]. Feng et al. [64] studied vacuum UV (VUV) as a feed pretreatment method to reduce membrane fouling. VUV offers the advantage of not needing oxidants or additional treatment chemicals. The experiment’s results showed that following VUV pretreatment, the removal of protein-like materials was enhanced by 40.1%, and the protein/polysaccharide ratio in the biofouling layer decreased from 34.6% to 15.8%, respectively. Furthermore, the richness and diversity of the bacterial community were decreased as a result of the VUV pretreatment.

Although UV disinfection offers several advantages, such as simplicity, lack of chemical additives, minimal space requirement, short contact time, and fewer harmful by-products, it is relatively costly, and its efficacy is limited when the microorganisms are capable of photo-reactivation [23,62,63].

### 5.2. Nutrient Limitation

Limiting biodegradable organic nutrients in the feed water, namely assimilable organic carbon (AOC), and phosphorous, is being studied as an approach to control unwanted biofilm growth on membranes. AOC concentration is of particular importance in controlling biofilm growth because AOC can be easily assimilated by bacteria and converted to cell mass. However, there is a disagreement in the literature as to the optimal AOC concentration to limit biomass growth; some studies proposed a concentration less than 100 µg/L [65]; whereas others suggested a concentration of 50 µg/L, which is similar to that of AOC in groundwater [66]. Several methods exist for reducing the AOC concentration in the feed water, including activated carbon adsorption, biological filtration, and sand filtration [23].

Limiting phosphorus levels in the feed water has also affected membrane biofouling. Phosphorous removal from feed water can be done through chemical precipitation, crystallization, ion exchange, and adsorption [23,67,68]. Generally, nutrient limitation studies monitor biofouling by independently restricting either carbon or phosphorous. Javier et al. [69] investigated the effect of the carbon-to-phosphorus ratio and the simultaneous restriction of both nutrients on membrane biofouling. The biofilm development was monitored when the phosphorous concentration was reduced to ≤0.3 μg P·L^−1^, along with two carbon concentrations (250 C L^−1^ and 30 μg C·L^−1^). The results demonstrated that phosphorous limitation delayed biofilm formation effectively when combined with low AOC concentration (a slower pressure drop increase was observed when 30 μg C·L^−1^ was used), and lower total cell counts (TCC) values were obtained, indicating reduced bacterial growth. In another study by Javier et al. [67], the cleanability of biofilms grown on RO membranes under low phosphorous concentrations (3 and 6 μg/L) was studied. The results showed that biofilms grown with a phosphorous concentration of 3 μg/L were easier to clean off hydraulically compared to biofilms grown at 6 μg/L. The ease of biofilm removal was attributed to the fact that at lower phosphorous concentrations, there are more soluble polymers in the EPS, which reduces the adhesive strength of biofilms.

### 5.3. Optimization of Feed Spacer Geometry and Hydrodynamic Conditions

Feed spacers separate membrane sheets, create flow channels, and promote turbulence in membrane processes [70,71]. Optimization of the hydrodynamic conditions can be achieved through the proper design of feed spacers. Generally, membrane fouling can be controlled by increasing the shear rate or turbulence near the membrane surface. The former can be increased by pumping the feed at a higher flow rate or by using thin flow channels above the membrane surface, while the latter can be enhanced by the appropriate design of feed spacers. In membrane biofouling, feed spacers provide a location where biofoulants can accumulate and ultimately spread to the membrane area. Moreover, high shear rates result in more compact biofilm structures, whereas increased turbulence enhances the transport of nutrients to the biofilm. Hence, feed spacer geometry is crucial for biofouling control. Lin et al. [72] investigated the effect of feed spacer geometry on spiral wound membrane (SWM) modules used in water treatment. The authors tested 16 feed spacers of varied geometries. The results showed that membrane biofouling depended on the variation of filament diameter, spacer thickness, and feed channel porosity. High-porosity feed spacer channels (>0.85) exhibited more biomass accumulation in the middle of spacer meshes because of lower shear stress. On the other hand, in low-porosity channels (<0.75), the biofilm developed from the region between spacer filaments and membrane surface, which leads to a larger area of dead zones and more severe biofouling. Their experiments showed that a channel porosity of 0.85 was optimum based on both hydraulic and anti-biofouling performance.

### 5.4. Membrane Cleaning

A decline in performance and operational parameters such as permeate flux, increase in TMP, and salt rejection are usually indicators that membrane cleaning is needed. Cleaning in biofouling control involves removing the accumulated biofoulants from the membrane in order to restore the permeate flux of a membrane. Cleaning methods are usually divided into physical and chemical cleaning methods [73]. Physical cleaning involves applying mechanical or hydraulic forces to remove foulants off the membrane surface, whereas, in chemical cleaning, the foulants are removed through the addition of chemical agents and are usually employed to get rid of irreversible fouling. Chemical cleaning can be conducted in situ, where the feed is replaced by the chemical cleaning agent, or ex situ, where the fouled membranes are removed and rinsed in tanks [28,74,75]. However, completely eradicating biofouling using chemical cleaning is ineffective, especially in more mature biofilms. Hence, frequent chemical cleaning cycles are required, which could shorten the life of the membrane. Moreover, combinations of the different cleaning methods can be used to achieve better results. A summary of the cleaning methods is presented in Table 5 [76,77].

### 5.5. Surface Modification

Surface modification refers to the alteration of membrane surface properties to make it more resistant to fouling and microorganism growth. Bacterial adhesion can be limited by reducing the roughness of the membrane surface, enhancing the hydrophilicity of the membrane, and/or inducing a negative charge on the membrane surface to repel negatively charged microorganisms. Moreover, biocides can suppress microorganisms’ growth and proliferation [25,77]. Anti-biofouling surface modification techniques include polymer blending, grafting, and coating [8,23,85,86].

#### 5.5.1. Polymer Blending

Polymer blending is a process in which two or more compounds are incorporated into the polymer solution during membrane preparation. The additives can be organic or inorganic materials. This technique alters the surface properties of membranes by modifying their bulk morphology [10,20,23].

Blended polymeric membranes with inorganic inclusions are referred to as mixed matrix membranes (MMM), and are characterized by improved thermal, physical, and mechanical properties. Several inorganic materials have been incorporated into the polymeric matrix of MMMs:Metal and metal oxide nanoparticles (NPs): Metal and metal oxide NPs, such as Titanium oxide (TiO_2_), silicon oxide (SiO_2_), and Zinc oxide (ZnO) exhibited excellent hydrophilicity and self-cleaning abilities when added to polymeric membranes. In addition, these NPs can generate free radicals and reactive oxygen species (ROS) and are able to interact with bacterial cells through electrostatic or van der Waals forces, disrupting the cellular membrane structure of microorganisms and inhibiting bacterial growth [20,87,88]. Kusworo et al. [89] doped polysulfone (PSF) membranes with TiO_2_ NPs. SEM images revealed that the addition of the NPs increased pore size. In addition, the hydrophilicity of the membrane was improved with the water contact angle decreasing from 61.83 to 41.67. The best pollutant removal was achieved with 1 wt% TiO_2_-PSF doped membranes. In another study, Aoudjit et al. [90] prepared and characterized a 10 wt.% TiO_2_/PVDF–TrFE nanocomposite membrane to separate Niflumic acid (NFA) from water. The photocatalytic activity of the incorporated TiO_2_ was tested and the results demonstrated a 91% NFA degradation efficiency after 6 h of solar irradiation at neutral pH. With respect to the reusability of the membrane, an efficiency loss of 9% was observed after three consecutive uses separated by cycles of washing with ultrapure water and drying in the sun. In addition, the authors found that the irradiation time was the most significant parameter affecting the performance of the nanocomposite membrane. Silver (Ag) NPs have also received a great deal of attention for their antibacterial properties and ability to reduce adhesion. The antibacterial properties of Ag NPs originate from the ability of released metal ions (Ag^+^) to interact with thiol (–SH) groups in microbial membrane cells; this interaction can deactivate certain proteins, which in turn causes the leakage of phospholipids and phosphate in cells, destroys cell DNA replication, and controls the propagation of microorganisms [25]. Spagnol et al. [91] immobilized AgNPs onto cellulose nanowhiskers (CWs) with polyvinyl alcohol (PVA) and poly (N-isopropylacrylamide) (PNIPAAm) as polymeric matrices, and their biological activity was evaluated against *Staphylococcus aureus* (*S. aureus*), *Bacillus Subtilis* (*B.subtilis*), *Escherichia coli* (*E. coli*), and *Candida albicans* (*C. albicans*). The properties of the films with CWSAc/AgNPs significantly influenced the antimicrobial activity displayed by each material, and all the films from PVA matrix exhibited the ability to inhibit bacterial growth.

Concerns regarding the incorporation of NPs into membranes include their leaching out into the retentate and/or permeate streams, which could compromise the safety of the treated water. To mitigate the risk of leaching, the modified membranes could be pre-washed in order to remove any free NPs lying on the surface of the membrane. Moreover, the agglomeration of NPs could interfere with the salt rejection capabilities of the membrane due to their uneven distribution on the membrane surface. Hence, further studies are needed to establish the safety, economic feasibility, as well as the applicability of NPs in large-scale operations [92].

Microporous materials: Microporous materials such as zeolites and metal-organic frameworks (MOFs) have high porosity and a large surface area that help increase the permeability, hydrophilicity, and anti-fouling behavior of membranes [25,93]. Beisl et al. [94] investigated the antibacterial activity of cellulose acetate/polyvinylpyrrolidone membranes coated with Ag NPs and cellulose acetate/silver ion-exchanged β-Zeolite membranes. The presence of silver ion-loaded zeolites improved the membrane hydrodynamic permeability by 56.3%; in addition, the silver ion-exchanged β-zeolite membrane showed complete *Escherichia coli* bacterial inactivation after just 210 min of contact time, for the same contact time, the Ag NPs incorporated membrane resulted in 99.95% reduction in bacterial activity indicating that both synthesized membranes possess strong bactericidal properties and are promising for biofouling mitigation. Dehghankar et al. [95] combined hydrophilic zirconium 1,4-dicarboxybenzene (UiO-66) and chromium (III) terephthalate (MIL-101) MOFs and faujasite (FAU) zeolites in a polyvinylidene fluoride (PVDF) polymeric matrix to study the anti-fouling properties of this MMM against bovine serum albumin (BSA). The best anti-fouling behavior was observed from the membrane containing 0.05 wt% UiO-66, 0.1 wt% MIL-101, and 0.1 wt% FAU, with a BSA rejection of 100% and 22.2% irreversible fouling.Hydrophilic polymers: Hydrophilic polymers (e.g., polyethylene glycol (PEG), polyethyleneimine (PEI), hyperbranched poly(amido amine) (PAMAM), polydopamine, and dendritic polyamide (PA)) are popular organic additives used to improve the anti-biofouling properties of membranes. Hydrophilic polymers possess a variety of polar groups capable of forming hydrogen bonds with water, which leads to improved membrane hydrophilicity and reduced microorganism adhesion [20,25,77]. In a study conducted by Ma et al. [96], the zwitterionic polymer poly(sulfobetaine methacrylate) (PSBMA) functionalized with graphene oxide (GO) nanocomposites (GO-PSBMA) was incorporated into a polyamide membrane (GO-PSBMA-1h). The synthesized membrane showed improved surface hydrophilicity, and a composition of 0.3 wt% GO-PSBMA-1h exhibited an 80% reduction in *Escherichia coli* attachment.

#### 5.5.2. Surface Grafting

Surface grafting refers to the addition of functional groups or charged species to the surface of the membrane. Grafting involves creating chemical bonds between the membrane surface and the grafted species. It is an easily implemented surface modification technique offering high chemical stability and control over the grafting density and spatial distribution; however, it is energy intensive and can be challenging to scale-up [20,23,75,97,98,99]. Surface grafting can be further divided into plasma- and photo-induced grafting. Membrane plasma treatment involves a polymerization reaction initiated by plasma-generated radicals, which are highly reactive when exposed to gaseous monomers or monomer solutions [20,23,100]. They react with the membrane’s monomers forming macromolecular chains that grow at the membrane surface. On the other hand, photo-initiated graft polymerization involves the formation of free radicals upon irradiation. Ultraviolet-induced graft polymerization is of particular interest due to its easy and controllable introduction of graft chains [20,23,97]. Numerous studies investigated the efficacy of surface grafting on membrane anti-biofouling behavior; for instance, Khongnakorn et al. [101] used plasma grafting polymerization with two different gases, i.e., argon (Ar) and carbon dioxide (CO_2_), to graft acrylic acid (AAc) on a cellulose triacetate (CTA) membrane. For both plasma gases, an increase in membrane hydrophilicity was noted (the decrease in wetting angle from 64.0° to 37.1° and 36.4° for CO_2_ and Ar plasma gases, respectively). Both plasma gases exhibited excellent anti-protein fouling properties; however, the Ar plasma-modified membranes provided more free radicals and showed better anti-fouling properties against proteins than polysaccharides. In another study, Vatanpour et al. [102] used UV irradiation to graft AAc on a polyamide membrane. The membrane was further modified through the incorporation of carboxylated multi-walled carbon nanotubes (COOH-MWCNTs). The membrane grafted with 50 g/L AAc under 5 min UV exposure showed the best filtration performance with a flux recovery ratio (FRR) of 80.2% during BSA filtration. The membrane embedded with 0.2 wt% COOH-MWCNTs showed the best water flux improvement (around 30%). All of the COOH-MWCNTs-modified membranes possessed excellent anti-fouling abilities with FRR values of 98–99%. However, membranes with higher concentrations of COOH-MWCNTs experienced extreme compaction of the grafting layer and were too stiff, which subsequently reduced the membrane surface hydrophilicity.

#### 5.5.3. Surface Coating

Surface coating, as the name implies, involves depositing a layer or a coat on the membrane surface through physical adsorption. The adsorbed coating can be stable or removable based on the adsorption affinity with the membrane surface. Stable coatings usually involve strong covalent bonding or the deposition of polyelectrolytes as thin films. Coatings held to the membrane surface by electrostatic interactions, weak van der Waals forces, or hydrogen bonding tend to be less stable and need to be replaced/re-applied with time [20,23,97]. Yuan et al. [103] coated a polydimethylsiloxane (PDMS) membrane with Ag NPs and Ag-MOFs. Both coated membranes exhibited enhanced hydrophilicity and anti-biofouling properties. However, the Ag-MOFs-coated membrane anti-biofouling performance was superior because it released silver ions in a slower, more controlled manner. Moreover, membrane autopsies showed that the surface of the Ag-MOFs-coated membrane had least amounts of proteins (0.004 mg/cm^2^), and polysaccharides (trace amounts). Falath et al. [104] conjugated a poly (vinyl alcohol) (PVA) membrane with Gum Arabic (GA). The results of this study showed that the membrane PVA-GA-5 containing 0.9 wt% GA enhanced the antibacterial properties by 98%, and chlorine resistance by 83%.

An interesting approach to membrane biofouling mitigation involves immobilizing enzymes on the membrane surface. Enzymes attached to the membrane surface can control biofouling through the degradation of EPS, its components (i.e., carbohydrates, proteins, lipids, etc.), or the destruction of the bacterial cell itself. For the degradation of polysaccharides, lyases and hydrolases are commonly used, whereas proteolytic enzymes (endopeptidase and exopeptidase) can be used to break down proteins. Enzymes that can degrade the EPS include proteinase K, trypsin, subtilisin, dispersin B, mutanase, dextranase, and antimycotic protein lysozyme [105,106,107]. Several studies have investigated the efficacy of immobilized enzymes in reducing membrane biofouling; for example, Tian et al. [108] deposited an antimicrobial lysozyme nanofilm on a polyamide membrane. Lysozymes are naturally occurring enzymes that function as antimicrobial agents by cleaving the peptidoglycan component of bacterial cell walls. The nanofilm was self-adhered on the membrane through an aqueous coating, and the modified membrane demonstrated a 50% reduction in bacterial activity compared to the unmodified membrane. Bao et al. [109] designed a hydrophilic, enzyme-immobilizing polymeric membrane. The first step of the membrane synthesis involved using radiation-induced graft polymerization to attach glycidyl methacrylate (GMA) to a polyethylene membrane (PE) sheet. In order to increase membrane hydrophilicity, the epoxy group in GMA was converted into dimethylamino-*γ*-butyric acid (DMGABA). Next, the enzyme acylase I (both in its active and inactive forms) was immobilized onto the DMGABA membrane, resulting in enzymatically active DMGABA (EI-DMGABA) and inactive DMGABA (Ina-EI-DMGABA) sheets. The conversion of GMA to DMGABA improved hydrophilicity, as seen in the reduction of water contact angle from 84.3° for GMA to 30.2° for DMGABA. With regard to biofilm formation, the biofilm formation ability of *Agrobacterium tumefaciens* was dramatically inhibited on EI-DMGABA, but not on the Ina-EI-DMGABA membrane. Mehrabi et al. [110] co-immobilized two enzymes, namely, α-Amylase and lysozyme, on a polydopamine/cyanuric chloride functionalized polyethersulfone membrane in an attempt to degrade *Staphylococcus aureus* and *Staphylococcus epidermidis* simultaneously. Interestingly, the action of the bienzymatic system removed more than 87% of the biofilm. An interesting study by Lan et al. [111] involved a regenerable pH-sensitive anti-biofouling system consisting of Proteinase-K-functionalized-PEGylated-silica (SPK) NPs. These smart enzymatic NPs generated an anti-biofouling layer on the membrane that is stable at pH 7.4 but can be released for regeneration at pH 10. Moreover, the generated anti-biofouling layer demonstrated excellent antibacterial activity against *Pseudomonas fluorescens* biofilms, and its activity lasted for at least 45 days. Figure 4 and Table 6 present a summary of studies involving the different surface modification techniques discussed above.

## 6. Future Prospects

Although there have been significant developments in the monitoring and control of biofouling, many issues still need to be addressed; for instance, the development of non-invasive, real-time monitoring systems of biofilms can provide fundamental information about the processes underlying biofilm formation [21]. Various sensors exist to investigate biofilm dynamics (i.e., oxygen levels, pH, and temperature), including optical, mechanical, and electrochemical sensors [121]. Moreover, measuring biofilm metabolites has emerged as an attractive approach to monitoring and gathering information about membrane biofilms. The metabolic activity can be assessed using various techniques, including NMR [122,123], Raman spectroscopy [124,125], bioluminescence, and fluorescence imaging [126,127,128]. Recently, there has been increasing research into biofilm-based biosensors, which employ biofilms as sensing elements in a detection device. Biofilm-based sensors usually employ bioluminescent and/or electroactive bacteria because they respond to stimuli by generating fluorescence and electric outputs without the need for further genetic modification. The attractiveness of these biofilm sensors lies in the fact that the biofilm immobilizes the bacteria without the need for chemicals; in addition, the EPS is extremely robust, which ensures long-term operation [121,129]. However, bioluminescent bacteria need to be in suspensions, which makes them unsuitable for real-time online monitoring; therefore, electroactive bacteria are better suited for such applications. Although biofilm-based sensors offer unique advantages over other real-time monitoring devices, there are still some issues that need to be addressed, such as that the bacterial response to different stimuli is complex and needs to be investigated thoroughly to ensure enhanced detection selectivity and sensitivity, low reproducibility of electroactive bacterial sensors, biofilm storage conditions, the size and portability of the devices, as well as high potential for contamination with other strains [121,129,130,131,132].

The energy consumption of water treatment processes is a huge obstacle, especially in developing countries and remote areas, partly due to the absence of a reliable electricity grid to power such technologies. For remote communities, small-scale desalination systems present a more sustainable economic alternative to trucking water. The lack of power or lack of grid reliability in such areas makes renewable energies a good alternative for power supply. Therefore, there has been growing interest in renewable energy-powered water treatment processes, in particular photovoltaics [133,134,135]. Monnot et al. [136] investigated the optimal configuration for a small-scale photovoltaic powered-RO plant to achieve the highest recovery rate and lowest costs. The parameters they considered included the type, number, and arrangement of the RO modules, stage configuration, the area of the photovoltaic panels, and the possibility of using evaporation ponds for concentrate disposal. The authors found that there was no ‘one-size-fits-all’ configuration, and that the optimal configuration was highly dependent on the water production needs. However, they were able to draw some useful conclusions; for example, a 65% recovery rate and cost reductions could be achieved with a double-stage RO configuration. Karavas et al. [137] compared five photovoltaic-powered SWRO designs: (1) without an energy storage device, (2) without a water storage device, (3) with both an energy and a water storage device, (4) with similar components to the third system with the addition of an FL-based energy management system (EMS), (5) and the last configuration had the same components as the third system but with a Fuzzy Cognitive Maps (FCM)-based ESM. Based on their techno-economical comparison, they found that the fifth configuration yielded the lowest cost and power loss. The optimum system reduced the cost of produced water per m^3^ by 22.78%, 12.53%, and 9.41%; in addition, the power losses by this configuration was cut by 58%, 54.15%, and 50.97%, compared to the first, second, and third proposed designs, respectively. One of the main challenges facing photovoltaics is that solar power is intermittent in nature; meaning that photovoltaic-powered desalination systems must operate intermittently with extended shutdown periods [138]. This intermittent operation mode leads to membrane fouling. Freire-Gormaly and Bilton [139] designed a photovoltaic-powered RO system in combination with a simulation model coupled to a GA-based algorithm. The developed system takes into account membrane fouling during the down periods. The study results showed that accounting for membrane fouling is essential because disregarding membrane fouling in the system design resulted in under-sized systems that could not meet the daily drinking water requirements.

Finally, carbon-based materials (CBMs), such as graphene, carbon nanotubes (CNTs), mesoporous carbon nanoparticles, and carbon quantum dots, have outstanding physical, chemical, thermal, electrical, and antibacterial properties that render them attractive for the development of high-performance membranes, with enhanced anti-biofouling properties. CBMs-modified membranes exhibit reduced biofilm adhesion and considerable antibacterial activity, particularly GO and CNTs modified membranes, due to their ability to cause physical disruption of the bacterial cell membrane, induce oxidative stress, and/or release of ROS. Although CBMs-modified membranes have exhibited promising potential in membrane biofouling mitigation, further studies are still required to achieve the commercial application level, particularly with regard to the large-scale production of these membranes, and their long-term durability and anti-biofouling performance. The latter is particularly challenging because, with extended operation, the EPS and dead microbial cells can condition the membrane surface facilitating subsequent adhesion of microbes, exacerbating biofouling effects. Therefore, a thorough understanding of long-term antibiofouling activity of CMBs-modified membranes is needed [140,141,142].

## 7. Conclusions

To conclude, membrane fouling, particularly membrane biofouling, is a major challenge to water treatment through membrane technologies. This paper presented the concept of membrane biofouling, its mechanism, the factors influencing biofouling-propensity, as well as the different tactics that can be used for membrane biofouling prediction, with a focus on AI techniques, and mitigation techniques. Based on the studies mentioned in this work, overcoming membrane biofouling requires the improvement of monitoring techniques especially noninvasive, nondestructive, real-time monitoring devices. Despite the successes reported on the modification of membranes to enhance their anti-biofouling properties, further research is needed on the safety, feasibility, and scalability of these methods, as most of the studies cited in this paper report on laboratory scale experiments, with little insight into the long-term behavior of the developed techniques under real operating conditions. Finally, the challenges facing the incorporation of AI-based predictive models need to be addressed to improve prediction accuracy and allow their translation into large-scale applications.

## Figures and Tables

**Figure 1 membranes-12-01271-f001:**
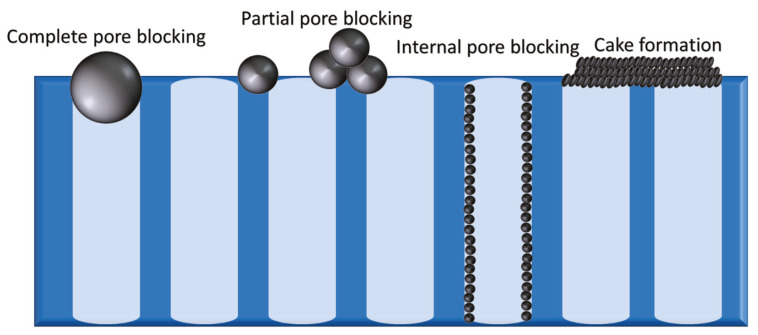
The four types of fouling (from **left** to **right**) complete pore blocking, partial pore blocking, internal pore blocking, and cake formation.

**Figure 2 membranes-12-01271-f002:**
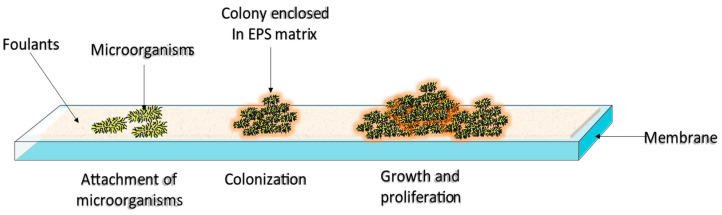
Biofouling mechanism.

**Figure 3 membranes-12-01271-f003:**
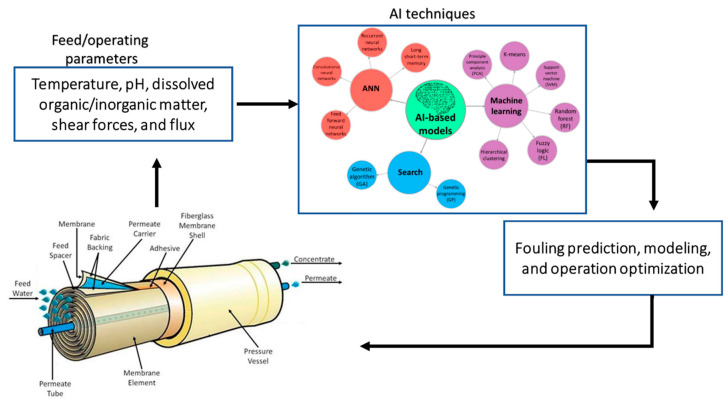
The use of artificial intelligence (AI) in membrane biofouling prediction.

**Figure 4 membranes-12-01271-f004:**
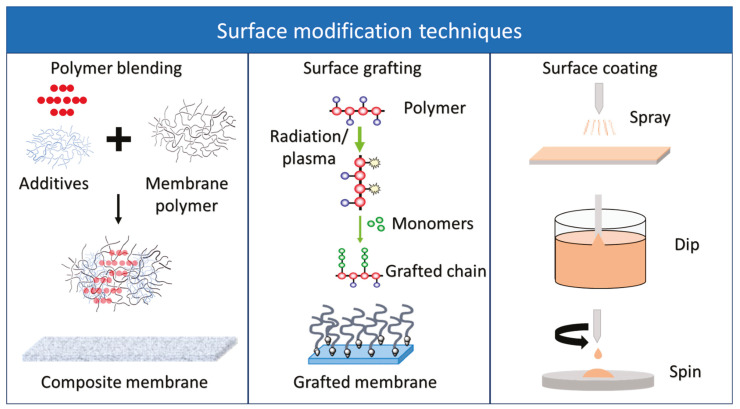
Membrane surface modification techniques.

**Table 1 membranes-12-01271-t001:** Membrane biofouling characterization techniques [5,23].

Element Characterized	Technique	Information Obtained
**Biofilm**	Epifluorescence microscopy (EFM)	-Morphological observations of biofilm
	Confocal laser scanning microscopy (CLSM)	-3D structure of the biofilm
	Electron microscopy (e.g., SEM and TEM)	-SEM enables imaging complex structures of biofilm -TEM enables the visualization of cross-sectional details of microorganisms -Mapping distribution of macromolecular subcomponents
	Atomic force microscopy (AFM)	-Biofilm surface topography
	X-ray microscopy	-Revealing the onset of bacterial colonization
	FTIR spectroscopy	-Analyzing microbial aggregates -Provides information about the chemical nature of foulants
	Nuclear magnetic resonance (NMR)	-Reveals the impact of biofouling on hydrodynamics and mass transport
**Microbial community**	Epifluorescence microscopy (EFM) with staining	-Microbial activity -Cell counts -2D distribution of bacteria in biofilm
	Confocal laser scanning microscopy (CLSM)	-3D structure of bacteria
	Heterotrophic plate counts (HPCs)	-Monitoring general bacteriological water quality
	Flow cytometry	-Species abundance and population dynamics
**Extracellular Polymeric Substance (EPS) matrix**	Phenol/sulfuric acid assay	-Carbohydrates quantification
	Lowry/Bicinchoninic acid (BCA) assay	-Protein quantification

**Table 2 membranes-12-01271-t002:** Membrane fouling models (adapted from [42,43,44]).

Fouling Models	Description	Governing Equation(s)
Resistance-in-series (RIS)	-Enable the determination of the fouling resistance form -Developed for dead-end MF -Vary with fouling mechanism (internal/external)	$J=\frac{\Delta P-\Delta \pi }{\mu R_{t}}$ $R_{t}=R_{m}+R_{cp}+R_{c}+R_{i}$
Pore blockage/Hermia’s models	-Describe the filtrate flux under constant pressure -Four blocking modes: complete pore blocking, standard blocking or pore constriction, intermediate pore blocking, and cake filtration	$\frac{d^{2}t}{dV^{2}}=k{\left(\frac{dt}{dV}\right)}^{n}$ $J=\frac{1}{A}\frac{dV}{dt}$
	Complete blocking; *n* = 2	$J=J_{0}\;\exp\left(-k_{b}t\right)$
	Standard blocking/pore constriction; *n* = 1.5	$J=\frac{J_{0}}{{\left(1+J_{0}{}^{0.5}k_{s}t\right)}^{2}}$
	Intermediate blocking; *n* = 1	$J=\frac{J_{0}}{1+J_{0}k_{i}t}$
	Cake filtration; *n* = 0	$J=\frac{J_{0}}{{\left(1+J_{0}{}^{2}k_{c}t\right)}^{0.5}}$
Combined cake filtration-pore blockage models	-Assume that the fouling occurs in three stages: pore constriction, pore blocking, and cake accumulation	QQ0=1(1+βQ0Cbt)2exp(−αCbJ0t1+βQ0Cbt)+∫0tαCbJ0(1+βQ0Cbtp)2exp(−(αCbJ0tp(1+βQ0Cbtp)))[(Rp0Rm)+(1+βQ0Cbtp)2]2+2(f′R′ΔpCbμRm2)(t−tp)dtp

**Table 5 membranes-12-01271-t005:** Comparison of chemical, physical, and nonconventional membrane cleaning methods (adapted from [10,23,74,75,77,78,79,80,81,82,83,84]).

Cleaning Method Category	Cleaning Method/Agent	Working Principle	Advantages	Disadvantages
Physical	Forward/reverse flushing	Pumping permeate water at high crossflow velocity through the feed side of the membrane (forward). Permeate flush direction alternated in forward and reverse directions (reverse).	1. Well-established method	1. Ineffective against irreversible fouling (e.g., pore-clogging with colloidal/dissolved materials)
	Backwashing	Negative pressure gradient is created across the membrane.	1. Easy implementation 2. Can be used with chemicals to enhance cleaning	1. Ineffective against irreversible fouling 2. Possibility of membrane damage
	Air flushing/sparging	Flushing along with air bubbles to create turbulence.	1. Easy to integrate into the membrane system 2. No chemicals involved 3. Low maintenance cost 4. Commonly combined with backwashing to remove biofoulants	1. Less efficient with hollow-fiber and SWMs 2. Air pumping cost is high
	Sponge ball	Sponge ball is used to scrub foulants from the membrane’s surface.	1. Used for heavily polluted membranes 2. Well-established method	1. Only applicable for tubular membrane modules
	Electrokinetics	The application of an electric field attracts particles from the membrane surface, damages cell membranes of microorganisms, and leads to the generation of oxidizing species.	1. Enhances efficiency of chemical cleaning	1. Mutagenic compounds may be created in water 2. Pretreatment required
Chemical	Acids (HCl, HNO_3_, H_3_PO_4_, H_2_SO_4_)	Suitable for removing inorganic foulants like salt precipitates or scales and metal oxides	1. Interfere with the weak electrostatic forces holding the microorganisms to the membrane	1. Frequent usage can damage the membrane 2. Need to be removed from the stream after cleaning
	Alkalis (NaOH, KOH, NH_4_OH)	Hydrolysis and solubilization of proteins and saccharides	1. Increase solubility of phenolic and carboxylic groups at high pH (~13) 2. Increase the negative charge of humic substances, hence weakening their bond with the membrane	1. Frequent usage can damage the membrane 2. Need to be removed from the stream after cleaning
	Surfactants	Solubilize foulants by enclosing them in micelles	1. Affect hydrophobic interactions with membrane, hence hindering biofilm formation	1. Frequent usage can damage the membrane 2. Need to be removed from the stream after cleaning
Nonconventional	Micro-nano bubbles (MNBs)	Foulant detachment due to shear stress generated by the collapse of MNBs, adsorption of foulant on MNBs surface due to hydrophobic interactions, and MNBs can generate hydroxyl radicals when they collapse, leading to the decomposition of organic foulants	1. Small size and large specific surface area 2. Extended residence time in solution 3. Environmentally friendly and non-chemical cleaning agents	1. Cavitation effects could lead to membrane damage 2. Cost and large-scale production need further study 3. Stability and storage issues need to be addressed
	CO_2_ nucleation	Combines hydraulic and chemical cleaning procedures, i.e., the formation of CO_2_ bubbles physically removes biofilms off of the membrane and the case a drop in pH acting as an acid-cleaning medium	1. CO_2_ gas is highly soluble in water 2. Formation of carbonic acid can facilitate removal of inorganic scaling	1. Technique still under research 2. Drop in pH may damage the membrane 3. ‘Green’ processes for obtaining CO_2_ are needed
	Ultrasound	US-induced cavitation minimizing foulants deposition and cell disruption	1. Chemical-free process 2. Can be combined with heat to improve cleaning 3. Membrane can be cleaned while in use 4. Hydroxyl and hydrogen peroxide radicals produced can act as disinfectants	1. Technique still under research 2. Scale-up of this technique still under study 3. May damage the membrane
	Hypersaline backwash	A high-concentration salt solution (hypersaline) is injected into the feed promoting direct osmosis across the membrane, while the reversible flow helps detach the biofilm and other foulants	1. On-line technique 2. High effectiveness 3. Ease of implementation 4. Chemical free	1. Technique still under research 2. Pulse concentration and timing need optimization
	Rhamnolipids	They act as cleaning agents (biosurfactants) that solubilize and remove the formed biofilms	1. Lower cost 2. Less toxic than conventional cleaning chemicals 3. Biologically produced	1. Technique still under research

**Table 6 membranes-12-01271-t006:** Surface modification techniques.

Method	Base Polymer Membrane	Modifier	Main Findings	Ref.
Polymer blending	Polysulfone	TiO_2_ NPs	-Improved hydrophilicity (wetting angle reduction from 61.83 to 41.67) -Increased pore size. -Best pollutant removal with 1 wt% TiO_2_ NPs dope PSF membranes.	[89]
	PVDF–TrFE	TiO_2_	-91% NFA photocatalytic efficiency was achieved after 6 h of solar irradiation at neutral pH.	[90]
	Polyvinylidene Difluoride	ZnO NPs	-Increased ZnO loading (from 2.5 to 7.5 wt%) improved membrane hydrophilicity. -ZnO incorporated membranes achieved BSA rejection of 93.4% ± 0.4 and flux recovery rate of 70.9% ± 2.1.	[112]
	Polyvinyl alcohol	SiO_2_ NPs	-SEM images showed that the SiO_2_ NPs and polymer matrix were compatible. -SiO_2_-modified membranes improved zinc ions removal to ~65%.	[113]
	Polyethersulfone	CuO/ZnO (CZN)	-Optimal CZN concentration was 0.2 wt% CZN. -SEM images showed the homogenous distribution of NPs in polymeric base. -BSA rejection was around 95% for all nanocomposite membranes.	[114]
	Polysulfone	MgFe_2_O_4_ and ZnFe_2_O_4_ NPs	-Membranes with 0.005 wt.% MgFe_2_O_4_ NPs exhibited the highest glucose rejection (96.52 ± 2.35%).	[115]
	Polyvinyl alcohol and poly (N-isopropyl acrylamide)	Ag NPs	-The type of polymeric matrix affected the antimicrobial activity. -PVA-based films exhibited the best antibacterial activity.	[91]
	Cellulose acetate/polyvinylpyrrolidone	Ag NPs and silver ion-exchanged β-zeolites	-Silver ion exchanged β-zeolites loaded membranes improved permeability by 56.3%, and increased salt rejection to 93%. -Silver ion exchanged β-zeolites loaded membranes showed the best antibacterial activity.	[94]
	Polysulfone	Chitosan–Ag NPs	-The modified membrane showed higher bactericidal properties (76% decrease in total cell count) and anti-adhesion capacity (60% less biofilm thickness and 75% less TOC compared to the unmodified membrane).	[116]
	Polyvinylidene fluoride	(UiO-66) and MIL-101 MOFs and FAU zeolites	-The optimal anti-fouling results were obtained for the 0.05 wt% UiO-66, 0.1 wt% MIL-101 MOFs, and 0.1 wt% FAU zeolite nanocrystals (~100% BSA rejection).	[95]
	Polyamide	GO-PSBMA	- The optimal additive concentration was 0.3 wt% GO-PSBMA. -Bacterial adhesion reduced by 80%.	[96]
	Sulfonated polysulfone (SPSf)	Tröger’s base (TB) polymer	-Blending enhanced surface and total porosity. -SPSf/TB blended membranes had slightly lower BSA retention (86.5–94.6%) than pristine SPSf membranes (94.7%).	[117]
	Polysulfone	Poly(methyl methacrylate-co-dimethyl aminoethyl methacrylate) (P(MMA-co-DMAEMA)) and 2-carboxyethyl acrylate	-The amount of adhered total *E. coli* on the membrane surface decreased in the following order for the different blended membranes: PSF > PSF-PMD > PSF-qPMD > PSF-nPMD. -Best antibiofouling behavior achieved by PSF-nPMD because its net charge was close to zero (no electrostatic attractions negatively charged *E. coli* bacteria).	[118]
Surface grafting	Polyamide	3-allyl-5,5-dimethylhydantoin (ADMH)	-The modified membrane showed improved microbial adsorption compared to the unmodified membrane.	[119]
	Cellulose triacetate (CTA)	Acrylic acid (AAc)	-Both plasma gases increased membrane hydrophilicity (water contact angle reduced from 64.0° to 37.1° and 36.4° for CO_2_ and Ar plasma gases, respectively). -The hydrophilicity increased due to the presence of hydrophilic functional groups such as carboxyl O–C=O and –COOH. -Ar gas generated more free radicals than CO_2_. -Anti-fouling behavior against proteins was better than polysaccharides.	[101]
	Polyamide	AAc and MWCNTs	-Membrane embedded with 0.2 wt% COOH-MWCNTs showed the best water flux improvement (around 30%). -Higher COOH-MWCNTs concentrations reduced the hydrophilicity of the membrane. -All of the COOH-MWCNTs-modified membranes possessed excellent anti-fouling abilities with FRR values of 98–99%.	[102]
	Polyvinylidene fluoride (PVDF)	Quaternary ammonium compounds (QACs) and silica NPs	-Approximately 99.9% bacterial inhibition was achieved with the modified membrane.	[120]
Surface coating	Poly (vinyl alcohol) (PVA)	Gum Arabic (GA)	-PVA-GA-5 containing 0.9 wt% GA enhanced the antibacterial properties by 98%, and chlorine resistance by 83%.	[104]
	Polydimethylsiloxane (PDMS)	Ag NPs and Ag-MOFs	-Both Ag NPs and Ag-MOFs coated membranes exhibited enhanced hydrophilicity and anti-biofouling properties. -The superior anti-biofouling performance of the Ag-MOFs-coated membrane was attributed to the slow and controlled release of silver ions. -The uncoated membrane had 14-times higher protein amounts than the Ag-MOFs-coated membrane (0.004 mg/cm^2^) on its surface.	[99]

## Data Availability

No new data were created or analyzed in this study. Data sharing is not applicable to this article.

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
