# Peer review of "A Review on Membrane Biofouling: Prediction, Characterization, and Mitigation"

_membranes, 2022, doi:10.3390/membranes12121271_

Round 1
Reviewer 1 Report
This review article with the title “ A review on Membrane Biofouling: Prediction, Characterization, and Mitigation” briefly explained the summary related to the biofouling mechanism in the membrane process, its identification, and mitigation techniques. Along with a significant explanation of future prospects in the combination with AI, particularly machine learning techniques. In my opinion, this review article is well explained and related to the journal publishing domain, and considered for publication with some minor comments
1. Although the review paper is well written, it would be suggested that if authors add the schematics of the membrane modification techniques in each case (like schematics for the polymer blending, surface grafting, etc.). It would increase the article's visibility and readability.
2. It is suggested to add the summary of various fouling models that can be used for the prediction of fouling models.
3. In some parts of the article there are some grammatical mistakes that which author needs to address
Author Response
Please find our attached responses

Reviewer 2 Report
please find attached

Author Response
Please find our attached response to the reviewers' comments.

Round 2
Reviewer 2 Report
The authors have addressed most of my comments satisfactorily. The manuscript in its present form is acceptable for publication.